# The Detection of Motor Bearing Fault with Maximal Overlap Discrete Wavelet Packet Transform and Teager Energy Adaptive Spectral Kurtosis

**DOI:** 10.3390/s21206895

**Published:** 2021-10-18

**Authors:** D.-M. Yang

**Affiliations:** Department of Mechanical and Automation Engineering, Kao-Yuan University, Kaohsiung 821, Taiwan; dmyang@cc.kyu.edu.tw

**Keywords:** bearing fault detection, maximal overlap discrete wavelet packet transform, Teager energy adaptive spectral kurtosis

## Abstract

Motor bearings are one of the most critical components in rotating machinery. Envelope demodulation analysis has been widely used to demodulate bearing vibration signals to extract bearing defect frequency components but one of the main challenges is to accurately locate the major fault-induced frequency band with a high signal-to-noise ratio (SNR) for demodulation. Hence, an enhanced fault detection method combining the maximal overlap discrete wavelet packet transform (MODWPT) and the Teager energy adaptive spectral kurtosis (TEASK) denoising algorithms is proposed for identifying the weak periodic impulses. The Teager energy power spectrum (TEPS) defines the sparse representation of the filtered signals of the MODWPT in the frequency domain via the Teager energy operator (TEO); the TEASK helps determine the most informative frequency band for demodulation. The methodology is compared in terms of performance with the fast Kurtogram and the Autogram methods. The simulation and practical application examples have shown that the proposed MODWPT-TEASK method outperforms the above two methods in diagnosing defects of motor bearings.

## 1. Introduction

Rolling element bearings are one of the most critical supporting components in rotating electrical machinery due to the fact that approximately 41–42% of induction machine failures arise from faulty bearings [1,2]. Hence, early fault detection and diagnosis of rolling bearings is to prevent unscheduled machine breakdown, reduce maintenance costs, improve productivity, and avoid catastrophic accidents.

Vibration signals measured from bearings usually contain rich information on the machine condition and have been widely used for bearing fault diagnosis [2]. When a defect appears on the bearing surface, the rolling elements pass over the defects and thus generate a series of periodic impulses to excite higher frequency bearing resonances during this process; however, the diagnostic information related to bearing faults is easily submerged in the heavy background noise and other vibration sources, increasing the difficulty of bearing fault detection.

When impacts induced by rolling bearings, they will excite the resonance of bearing elements and housing structure, which causes the modulation of vibration signals in the high frequency bands. Envelope demodulation has been widely used to demodulate vibration signals to extract bearing characteristic defect frequencies, which strongly require vibration data with high signal-to-noise ratio (SNR) [3,4]. To enhance the SNR and accentuate fault features, a band-pass filter is usually set manually around the desired resonant frequency band before demodulation is performed; however, the main challenge in envelope demodulation is to find a suitable frequency band for amplitude demodulation [5]. In practice, however, the fault-induced features are extremely weak and often buried in strong background noise at an early stage. Under this situation, the traditional envelope analysis is unable to eliminate the undesired noise contained in the signal and eventually results in the failure of the early stage fault detection [6].

To address this issue, Duan et al. [5] proposed an multi-band envelope spectral extraction technique to determine the optimal frequency band for narrowband demodulation in bearing fault diagnosis; however, it is known that the narrowband signal demodulation is easily affected by the independent fault transient components and background noise. To mitigate this problem of background noise and random phase noise caused by bearing slippage, Xu et al. [6] applied an autocorrelated envelope method to identify the bearing faults at a very low SNR.

Recently, numerous methods have been proposed to filter the signal for noise reduction before envelope analysis to improve the SNR. One of the first systematic approaches based on kurtosis to find the optimal frequency band before envelope demodulation was undertaken by Antoni and Randall [7,8]. The Kurtogram displays the values of spectral kurtosis (SK) in a two-dimensional diagram as a function of the central frequency and the bandwidth window. The Kurtogram is able to automatically identify frequency bands of the signal corresponding to highly impulsive components for demodulation without prior knowledge of a signal. Nevertheless, the Kurtogram based on the short-time Fourier transform (STFT) was found impractical due to its computational complexity. To reduce computing time, Antoni [9] then proposed the fast Kurtogram (FK) to automatically find the optimal frequency band according to the maximum value of the kurtosis of the filtered time signal in different filter banks for on-line fault detection. Since that time, the FK has been the most widely used band selection tool and become a benchmark for bearing fault diagnosis [10]. Because the kurtosis calculated in the time domain is very sensitive to the noise component and the non-periodic transient impulse in the signal, the FK may yield the incorrect or suboptimal frequency band in detecting bearing-fault induced transients under a low SNR and non-Gaussian noise [11,12].

In order to obtain a higher SNR and enhance fault features in the presence of strong background noise, a number of improved Kurtogram-based methods have been proposed by many researchers. For example, Barszcz and Jablonski [11] developed the Protrugram to characterize the cyclostationarity of repetitive transients in the presence of non-Gaussian noise by calculating the kurtosis of the envelope spectrum. For the first time, the Protrugram method made a major shift by measuring statistics from the time domain to the frequency domain for detecting repetitive transients; however, the required bandwidth in the Protrugram method has to be preset. For a fixed bandwidth, the fault information may be difficult to extract from the noise, and thus the envelope spectrum may not reveal the fault characteristic frequencies. Then, Wang et al. [13] further proposed the enhanced Kurtogram by calculating the kurtosis of the envelope power spectrum of different nodes in wavelet packet transform (WPT) to determine the optimal frequency band for showing fault characteristic frequencies. Later, by extension of the concept of the Protrugram, Antoni [12] proposed the infogram to measure the negentropy of both the squared envelope (SE) in the time domain and squared envelope spectrum (SES) in the frequency domain of the signal to detect repetitive transients and periodic components in the presence of strong impulsive noise, respectively. To bypass electromagnetic interference noise on the signals, Smith et al. [14] studied the optimized spectral kurtosis (OSK) to select the best demodulation band with the maximum kurtosis by minimizing the bandwidth to maximize the SNR. Soon after, Moshrefzadeh and Fasana proposed the Autogram [15], which utilizes the autocovariance function of the second-order cyclostationary signal to calculate the kurtosis of the unbiased autocorrelation of the squared envelope of the demodulated signals. Based on the maximum kurtosis to select the optimal band for demodulation, the Autogram can enhance the fault-related periodic components by reducing the effects of non-periodic impulses and noise. Xu et al. [16] proposed the PMFSgram by calculating kurtosis based on only part of the SES of the band filtered signals instead of the entire of SES to avoid inference from random undesired impulses in SES, which can select the optimal frequency band with more useful information and less noise for bearing fault diagnosis. Liang et al. [4] introduced the ALKurtogram by calculating the averaged local kurtosis to measure both the local and overall impulsiveness of bearing signals to further improve the performance of the original FK. Xu et al. [17] proposed a method combining singular value decomposition and the composite SES to denoise the vibration signal and separate impact series for bearing fault diagnosis. To eliminate the interference of non-Gaussian noise and extract periodic transients from noisy signals, Wang et al. [18] presented a method, called the sub-band averaging Kurtogram, which computes the kurtosis of sub-bands obtained by the dual-tree complex wavelet packet transform of each sub-signal. Then, the optimal frequency band is selected for the envelope analysis based on average kurtosis of corresponding sub-bands. As seen from the above, the combination of kurtosis-based-indexes and envelope analysis has become one of the most popular procedures for the diagnosis of incipient faults in rolling element bearings. The relationship among kurtosis-based-indexes, the envelope analysis, and the SSE is discussed in Ref. [19].

Although the Hilbert transform (HT) has been the most widely used for bearing fault demodulation in recent years, the HT of noise-corrupted signals usually produces spurious amplitudes at negative frequency [20]. Moreover, the HT has low adaptability to the local signal structure and the energy leakage may limit the demodulation ability [21]. In the preliminary study [22], we have been unable to detect the weak bearing fault characteristics using the HT-based SES. To overcome this problem, the Teager energy operator (TEO) is adopted in this study because the TEO is an effective envelope demodulation method [21] and has a good noise suppression capability to extract weak fault features of rolling bearings [23]. Unlike the Hilbert envelope demodulation method, the TEO can directly compute the signals without a band-pass filtering process and the computation of the TEO is simpler and faster than band-pass filtering [24]. Moreover, the TEO can extract the information of amplitude modulation (AM) and frequency modulation (FM) caused by the fault impulses by enhancing the SNR and then the Teager energy spectrum (TES) can detect the fault frequency in the presence of noise and interferences [25].

It has been apparent from the studies mentioned above that these SK-based techniques are capable of yielding valid results under certain conditions but some cases in which these methods are not always effective in identifying the bearing optimal frequency band due to the inclusion of other components, such as strong background noise, interference from other vibration components, and non-stationary slippage caused by rolling elements. For the further improvement in the accuracy and efficiency of the algorithm in determining optimal frequency bands and implementing demodulation, this study introduces a new indicator named the TEASK for bearings diagnostics. The TEASK includes the following processes: the MODWPT was first used to decompose the vibration signal for noise reduction, then each sub-band after decomposition was demodulated by the TEPS via TEO. Finally, the kurtosis values of the overlapped and segmented TEPS sequences were calculated to provide a quantized TEASK profile for each frequency band with a bandwidth and a center frequency. The methodologies were validated on simulated and experimental data. Furthermore, the performance of the methodology was compared with the FK and the Autogram methods for analysis of the same signals.

The novelty and main contributions of this paper can be summarized as follows:(1)By integrating the MODWPT with TEPS, a new bearing detection tool, named the MODWPT-TEASK, for the analysis of vibration signals resulting from bearings with localized defects is presented, which can be used for the diagnosis of single and early weak faults.(2)The MODWPT-based sub-band analysis is used to analyze the bearing vibration signals and extracts the characteristic features of various bearing faults. The sparse representations of fault features of small incipient bearing defects are obtained by calculating the TEPS of each sub-band signal at different decomposition levels via TEO, thereby enhancing the accuracy of bearing fault detection.(3)The TEASK dynamically measures the protrusion of the sparse representation to highlight bearing fault characteristic frequency and its harmonics, thereby providing accurate detection results.(4)The proposed fault detection method based on the MODWPT-TEASK can effectively detect the very weak fault characteristics of bearings under very low SNR. The MODWPT-TEASK also detects bearing fault more accurately than two state-of-the-art methods (FK and Autogram).

The rest of the paper is outlined as follows. In Section 2, the background theory needed for the application of the proposed method is briefly introduced and then the proposed methodology is presented. Numerical simulations were conducted to evaluate the proposed methodology and the comparisons with the FK and the Autogram methods are given in Section 3. Section 4 describes the experimental rig. In Section 5, experimental validation using actual vibration data is presented and the results are compared with the FK and the Autogram methods to examine the performance of the proposed method. Finally, conclusions are provided in Section 6.

## 2. Proposed Method

### 2.1. Maximum Overlap Discrete Wavelet Package Transform

The MODWPT is an enhanced version of the discrete wavelet transform (DWT). Compared to the DWT, the MODWPT is undecimated and each decomposition level has the same number of wavelet coefficients, thus ensuring that it contains all the essential bearing vibration information and possessing the same circular shift properties to maintain a consistent cycle time between two consecutive fault-induced impulses at different nodes. Moreover, the MODWPT is an energy-conserving transformation, that is, the sum of the total energies of the MODWPT coefficients equals the energy of the signal. The MODWPT algorithm is briefly described below, while the detailed information can be found in [26,27].

Let us consider a discrete time sequence $X=\{x_{0},x_{1},\ldots ,x_{N-1}\}$ of N samples obtained by sampling a continuous time signal x(t) with a sampling frequency $F_{s}$. The MODWPT can be obtained by convolving X with a subset of the MODWPT filters. The low-pass scaling filter can be expressed by $\left\{g_{m}:m=0,1,\ldots ,M-1\right\}$ and its quadratic mirror high-pass wavelet filter can be by $\left\{h_{m}:m=0,1,\ldots ,M-1\right\}$, where *M* is the length of filter and M≤N. The filters $g_{m}$ and $h_{m}$ are related to each other as
(1)$$h_{m}={\left(-1\right)}^{m}g_{M-m-1}\;\mathrm{or}\;g_{m}={\left(-1\right)}^{m}h_{M-m-1}$$

A scaling of the MODWPT filters is required to conserve energy and their filters can be given by ${\tilde{g}}_{m}=g_{m}/\sqrt{2}$ and ${\tilde{h}}_{m}=h_{m}/\sqrt{2}$. The transfer functions, $\tilde{G}\left(f\right)$ and $\tilde{H}\left(f\right)$ corresponding to ${\tilde{g}}_{m}$ and ${\tilde{h}}_{m}$, respectively, are given by
(2)$$\begin{array}{cc} \tilde{G}\left(f\right) & =\sum_{m=0}^{M-1}{\tilde{g}}_{m}e^{-j2\pi fm} \end{array}$$
(3)$$\begin{array}{cc} \tilde{H}\left(f\right) & =\sum_{m=0}^{M-1}{\tilde{h}}_{m}e^{-j2\pi fm} \end{array}$$

At the first level, j=1, $W_{0,0}\left(=X\right)$ is circularly filtered by $\tilde{H}\left(f\right)$ and $\tilde{G}\left(f\right)$ to obtain correspondingly, the first level coefficients $W_{1,0}=\left\{W_{1,0,k}:k=0,1,\ldots ,N-1\right\}$ and $W_{1,1}=\left\{W_{1,1,k}:k=0,1,\ldots ,N-1\right\}$. Next the filters ${\tilde{h}}_{m}$ and ${\tilde{g}}_{m}$ of the second level (j=2) are added one zero between the elements of filter coefficients to give ${\tilde{g}}_{m}=\left\{{\tilde{g}}_{0},0,{\tilde{g}}_{1},0,\ldots ,{\tilde{g}}_{M-2},0,{\tilde{g}}_{M-1}\right\}$ and ${\tilde{h}}_{m}=\left\{{\tilde{h}}_{0},0,{\tilde{h}}_{1},0,\ldots ,{\tilde{h}}_{M-2},0,{\tilde{h}}_{M-1}\right\}$, respectively. Moreover, their corresponding transfer functions are given by $\tilde{H}\left(2f\right)$ and $\tilde{G}\left(2f\right)$. Now $W_{1,0}$, and $W_{1,1}$ are circularly filtered by $\tilde{H}\left(2f\right)$ and $\tilde{G}\left(2f\right)$ to give the second level vectors $W_{2,0}$, $W_{2,1}$, $W_{2,2}$ and $W_{2,3}$. For subsequent levels *j* of the transform, $2^{j-1}-1$ zeros are inserted between the elements of ${\tilde{h}}_{m}$ and ${\tilde{g}}_{m}$ whose transfer functions are given by $\tilde{H}\left(2^{j-1}f\right)$ and $\tilde{G}\left(2^{j-1}f\right)$ and the filtering process continues until the desired level $J_{o}$. The process of filtering up to level j=4 is illustrated in Figure 1.

From above recursive process, the MODWPT coefficients corresponding to each node (j,n), $W_{j,n}$, can be calculated as:(4)$$W_{j,n,k}=\sum_{m=0}^{M-1}{\tilde{r}}_{n,m}W_{j-1,\left[\frac{n}{2}\right],\left(k-2^{(j-1)m}\right)\;\mathrm{mod}\;N}$$
where
$$\begin{array}{cc} {\tilde{r}}_{n,m} & =\;\;\left\{\begin{array}{cc} {\tilde{q}}_{m}, & \;\mathrm{if}\;n\;\mathrm{mod}\;4=0\;\mathrm{or}\;3\\ {\tilde{h}}_{m}, & \;\mathrm{if}\;n\;\mathrm{mod}\;4=1\;\mathrm{or}\;2 \end{array}\right. \end{array}$$
where *n* ($n=0,1,2,\ldots ,2^{j}-1$) represents frequency band number under each transform level, *j* and the "mod" denotes the modulus after division.

### 2.2. Teager Energy Operator

The TEO is a nonlinear differential operator and can be used to track the amplitude envelope and instantaneous frequency of narrowband signals [28,29]. The TEO for the continuous signal s(t) is defined as:(5)$$\Psi_{c}\left[s\left(t\right)\right]={\left[\frac{ds(t)}{dt}\right]}^{2}-s\left(t\right)\cdot \frac{d^{2}s\left(t\right)}{dt^{2}}={\left[\dot{s}\left(t\right)\right]}^{2}-s\left(t\right)\ddot{s}\left(t\right)$$
where $\dot{s}\left(t\right)$ and $\ddot{s}\left(t\right)$ represent the first and second derivatives of s(t), respectively.

For discrete-time signal s(n), the difference equation is used instead of the differential equation:(6)$$\Psi_{d}\left[s\left(n\right)\right]={\left[s\left(n\right)\right]}^{2}-s\left(n-1\right)s\left(n+1\right)$$

An important property of the TEO in discrete time is that the signal energy at time instant *n* can be calculated by only three samples s[n−1], s[n] and s[n+1].

Now consider an arbitrary signal $s_{a}\left(t\right)$
(7)$$s_{a}\left(t\right)=a\left(t\right)\cos\left(2\pi f_{a}\left(t\right)t+\theta \right)$$
where a(t) is the time-varying amplitude, $f_{a}\left(t\right)$ is the time-varying instantaneous frequency of signal $s_{a}\left(t\right)$ and θ is an initial phase.

The signal $s_{a}\left(t\right)$ of the energy operator is given as [30]
(8)$$\begin{array}{c} \Psi_{c}\left[s_{a}\left(t\right)\right]=4\pi^{2}a^{2}\left(t\right)f_{a}^{2}\left(t\right) \end{array}$$
(9)$$\begin{array}{c} \Psi_{c}\left[{\dot{s}}_{a}\left(t\right)\right]=16\pi^{4}a^{2}\left(t\right)f_{a}^{4}\left(t\right) \end{array}$$

By combining Equations (Equation 8) and (Equation 9), the absolute value of time-varying amplitude envelope a(t) and the instantaneous frequency $f_{a}\left(t\right)$ can be estimated as: (10)$$\begin{array}{c} \left|a\left(t\right)\right|=\frac{\Psi_{c}\left[s_{a}\left(t\right)\right]}{\sqrt{\Psi_{c}\left[{\dot{s}}_{a}\left(t\right)\right]}} \end{array}$$(11)$$\begin{array}{c} f_{i}\left(t\right)=\frac{1}{2\pi }\frac{\Psi_{c}\left[s_{a}\left(t\right)\right]}{\sqrt{\Psi_{c}\left[{\dot{s}}_{a}\left(t\right)\right]}} \end{array}$$

It can be seen from Equation (Equation 8) that the most significant element of the TEO is the square product of the instantaneous amplitude and the instantaneous frequency of a signal. The envelope spectrum of the proposed approach is calculated by using these unique characteristic features of the TEO.

### 2.3. Teager Energy Adaptive Spectral Kurtosis

To take advantage of the capabilities provided by the TEO, this technique is used to demodulate each wavelet coefficient, $W_{j,n,k}$, instead of the HT. The TEO has also good localization characteristics to enhance the repetitive transients and increase the demodulation accuracy [21,31]. Then the output of TEO for $W_{j,n,k}$ is given by
(12)$$T_{j,n}\left(k\right)=\Psi \left[W_{j,n,k}\right]$$

According to the abilities to extract fault signatures, the TES outperforms the envelope spectrum [32]; therefore, the signal $T_{j,n}\left(k\right)$ is transformed to the frequency domain using L-point discrete Fourier transform (DFT) and its TES can be expressed as
(13)$$F_{j,n}\left(\nu \right)=\frac{1}{L}\left|\sum_{k=0}^{N-1}T_{j,n}\left(k\right)e^{-j2\pi k\nu /L}\right|,\;\nu =0,1,2,\cdots ,L-1$$

It is demonstrated that the envelope power spectrum of the signals can provide a good sparse representation of bearing fault signals and enhance the fault characteristic frequency effectively [13,33]. According to power spectral estimation techniques, the power spectrum can be obtained by either parametric or non-parametric approaches [34]. From a practical point of view, the direct approach is the most widely used because of the efficiency of the fast Fourier transform (FFT). In the direct approach, the TEPS of each node, denoted by $P_{j,n}\left(\nu \right)$, is calculated as the magnitude squared of $F_{j,n}\left(\nu \right)$:(14)$$P_{j,n}\left(\nu \right)={\left|F_{j,n}\left(\nu \right)\right|}^{2}$$

Then a rectangular sliding window w(n) is used to window $P_{j,n}\left(\nu \right)$ to truncate the frequency series so that the $r^{\mathrm{th}}$ windowed frequency band segmentation is given by
(15)$$P_{j,n}^{r}\left(\nu \right)=P_{j,n}\left(\nu +rQ\right)w\left(\nu \right)$$
where *Q* is the shift samples at a time under the window w(ν) and Q≤L. The rectangular window function is defined as
(16)$$\begin{array}{cc} w(n) & =\left\{\begin{array}{cc} 1, & 0\leq n\leq N_{w}\\ 0, & \mathrm{else} \end{array}\right. \end{array}$$
where $N_{w}$ represents the length of the window.

Moreover, the process of merging the overlapped and shifted frequency segments can be expressed as
(17)$${\hat{P}}_{j,n}^{r}\left(\nu \right)=\left(1-\frac{\nu }{L}\right)P_{j,n}^{r}\left(\nu \right)+\frac{\nu }{L}P_{j,n}^{r+1}\left(\nu \right),\;r=1,2,\cdots ,R$$
where *R* is the number of segments and the largest integer satisfies $R\leq (\frac{L}{2}-N_{w})/Q$.

The spectral kurtosis (SK) is a spectral statistical indicator which can detect not only the transient impulses in the presence of strong background noise but also reveal the impulsive components in frequency domain [7,8]. Moreover, the kurtosis of the TES can well reflect periodic impulse signal with a low SNR and is not sensitive to non-periodic transient impulse component [35]. In Ref. [36], an adaptive SK can optimize filter bandwidth and locate center frequency to easily identify the resonance frequency band; therefore, the proposed indicator, the TEASK, is to integrate the merits of adaptive SK and the TEPS to identify the optimal resonance frequency band. For the $r^{\mathrm{th}}$ narrow band, the TEASK of ${\hat{P}}_{j,n}^{r}\left(\nu \right)$ is defined as
(18)$$K^{r}\left(j,n\right)=\frac{\left(1/L\right)\sum_{\nu =0}^{L-1}{\left|{\hat{P}}_{j,n}^{r}\left(\nu \right)-{\bar{P}}_{j,n}^{r}\left(\nu \right)\right|}_{+}^{4}}{\{\left(1/L\right)\sum_{\nu =0}^{L-1}|{\hat{P}}_{j,n}^{r}\left(\nu \right)-{\bar{P}}_{j,n}^{r}\left(\nu \right){{|}_{+}^{2}\}}^{2}}$$
where the operator ${\left|\cdot \right|}_{+}$ represents that only positive values are allowed while the values of the other data samples are set to zero. The ${\bar{P}}_{j,n}^{r}\left(\nu \right)$ is defined as
(19)$${\bar{P}}_{j,n}^{r}\left(\nu \right)=\frac{1}{N_{W}}\sum_{q=\nu }^{\nu +N_{w}-1}{\hat{P}}_{j,n}^{r}\left(\nu \right)$$

It is noted that the ${\bar{P}}_{j,n}^{r}\left(\nu \right)$ is the moving mean value of the TEPS and is used to adaptively determine the threshold level for characterizing the cyclostationarity of repetitive transients. As can be seen in Equation (Equation 18), the SK value is derived from a sliding spectral window ${\hat{P}}_{j,n}^{r}\left(\nu \right)$ to determine the center frequency and bandwidth via overlapping consecutive windows. By doing so, we can avoid some bands with high peak values of noise or the selected signal just in one frequency band, which may lead to inaccurate diagnostic results.

### 2.4. The Proposed Bearing Fault Detection Scheme

The flowchart of the proposed method is shown in Figure 2 and the details of each step are described below.

Preprocess the digitisation of the original vibration signal measured by an accelerometer to obtain the normalized signal x(k) with zero mean and unity standard deviation.Decompose x(k) at level *J* using Equation (Equation 4).Calculate the TEO of each node of the MODWPT using Equation (Equation 12).Apply the DFT to the squared TEO output to obtain the TES using Equation (Equation 13).Perform the direct approach to compute the TEPS according to the squared magnitude of TES using Equation (Equation 14).Calculate the TEASK values of all nodes using Equation (Equation 18).Represent the TEASK values of each node in a two-dimensional color map.Select the optimal frequency band $\left(j^{*},n^{*}\right)$ based on the overall highest TEASK value.Calculate the optimal TEPS $P_{j^{*},n^{*}}^{r}\left(\nu \right)$ using Equation (Equation 14).Identify bearing characteristic frequency and its harmonics.

## 3. Simulation Analysis

To verify the proposed method in detecting transient signals of a defective bearing, a vibration signal x(t) composed of periodic bursts of exponentially decaying ringing and white noise was used to simulate a defected bearing. The simulated signal from [37] was simply modified and its formula is given as
(20)$$\begin{array}{cc} x(t)= & \sum_{r}\sum_{k}e^{-\beta \left(t-rF_{s}/f_{d}-\tau_{r}\right)/F_{s}}\times \\ \; & \left(A_{k}\sin\left(2\pi f_{k}\left(t-rF_{s}/f_{d}-\tau_{r}\right)/F_{s}\right)\right)+n\left(t\right) \end{array}$$
where β is the attenuation factor, $F_{s}$ is the sampling frequency, $f_{d}$ is the fault characteristic frequency, $\tau_{r}$ represents the effect of random slippage of the rolling elements and a discrete uniform distribution is used to simulate the variation of $\tau_{r}$ caused by slippage with respect to $f_{d}$, $A_{k}$ represents the $k^{th}$ impact amplitude, $f_{k}$ denotes the $k^{th}$ resonant frequency and n(t) is a Gaussian distribution white noise.

In the simulation, the sampling frequency $F_{s}$ of a simulated signal with two resonant frequencies is taken as 12 kHz with 16,384 samples and a Gaussian white noise were added to obtain a noise-contaminated signal with a low SNR of −32 dB. All the parameters used in Equation (Equation 20) are given in Table 1.

The simulated signal is shown in Figure 3a. It can be clearly seen from the time-domain waveform that the simulated impulses are completely submerged in heavy noise. For clarity, Figure 3b displays a portion of periodic impulses with different slippages; its corresponding spectrum of Figure 3a is shown in Figure 3c, where it can be seen that the first resonance frequency band of 2500 Hz is almost overwhelmed by noise but the second resonance frequency band of 4100 Hz totally dominates the overall spectrum. From this raw spectrum, to obtain diagnostic information is difficult because the low harmonics of the periodic impulses have very low magnitudes and are easily masked by other component in the spectrum.

In order to determine the optimum demodulation frequency band and reveal the fault characteristic frequency 105 Hz and its harmonics, the FK proposed by Antoni [9] was used to analyze the same extremely noisy bearing fault signal because this technique has recently become an effective tool in detecting impulses from signals even buried in strong noise or interfered by other vibration sources [10]. The paving of the FK is presented in Figure 4a, where the node with maximum kurtosis value is considered to be the optimal node and is highlighted by a black dash-line rectangle. The optimal filter is at level 1.5 with the center frequency $f_{c}$ of 1000 Hz and the bandwidth Bw of 2000 Hz. The SES estimated after filtering of the noisy signal by using the above-mentioned band is presented in Figure 4b. Analyzing the filtered SES based on the FK, it is found that the FK failed to identify the fault characteristic frequency of 105 Hz. This fact indicates that the selected frequency band with maximum kurtosis in the time domain using the FK approach is greatly influenced by heavy background noise and the optimal frequency band should be selected through other approaches to extract the repeating frequency of periodic impulses.

Different from the FK, the kurtosis of the Autogram [15] is calculated according to the unbiased autocorrelation of the squared envelope of the demodulated signals. The diagnostic results from the Autogram-based combined SES are shown in Figure 4c,d, where the harmonics of the defect frequency 105 Hz are also not detected. The Autogram-based combined SES is selected here for comparison because this indicator can improve the fault detection ability of the original Autogram by summing the frequency bands with kurtosis larger than a certain threshold instead of only one frequency band with the highest kurtosis value for demodulation.

In order to overcome the above-mentioned problems, the proposed method was used to analyze the same simulated signal. Before performing the MODWPT on the signal, the most crucial step is the selection of the mother wavelet function as well as the decomposition level of the signal. Among orthogonal wavelets, Daubechies wavelets have been widely used because they match the transient components in vibration signals. Daubechies wavelets (db12) used in [15,38] were adopted in this study to decompose the signal and four decomposition levels was used in this investigation. Then the proposed method was implemented using the MODWPT along with the TEASK calculation of each frequency band signal via the TEPS to obtain the optimal band-pass filter which is shown in Figure 4e. At the same time, the window size used in the TEPS was set to half of the original signal with an overlapping of 50% (see p. 144 of [39] and p. 347 of [40]), which is half the window length for decreasing the TEPS estimate variance and then improving the SNR of the TEPS. It is shown that the node(3,6) indicated by the black dash-line rectangle with the maximum value of TEASK is considered to be the most informative node for identifying bearing fault signatures. The center frequency and its associated bandwidth are 4125 Hz and 750 Hz, respectively. It is interesting to note that the optimal center frequency 4125 Hz identified by the TEASK is quite well matched to the preset resonance frequency band of 4100 Hz, shown in Figure 3c. By further demodulation of this sub-band signal with Teager-energy spectral technique, the TEPS of the signal extracted from the node(3,6) by the MODWPT is shown in Figure 4f, where the defect characteristic frequency and its two harmonics are clearly visible.

The above simulation example indicates that the proposed method achieves higher performance, allowing to effectively extract fault signatures with a low SNR, when compared to the FK and the Autogram methods.

## 4. Experimental Apparatus

To verify the effectiveness of the proposed method in practical applications, we established a simplified bearing test rig, as shown in Figure 5. The experimental setup consists of a 2.2 kW 4-pole 3-phase induction motor mechanically coupled with a 5 kW DC generator via a torque sensor. The energy generated by the motor is absorbed by the generator. In order to investigate the various bearing faults under different loads, the drive-end bearing was replaced with two different kind of test bearing conditions.

The type of test bearing is NTN 6206ZZ, and parameters of the bearings are listed in Table 2. Spalls were introduced by electric discharge on the inner-race and the outer-race, respectively. The defect has a depth of about 0.5 mm and a diameter of about 1.0 mm. Each bearing was tested for the motor being unloaded (0%), 25%, 50%, 75% loaded, and fully loaded (100%). The data set of the bearing was acquired from the experimental system under three different operating conditions: (1) normal condition, (2) with outer-race defect, and (3) with inner-race defect; therefore, the vibration signals were acquired under different operating loads and bearing conditions. Sampling frequency of the data acquisition system is 12,000 Hz. In this experiment, each record of 16,384 samples was recorded under five different motor load conditions; two hundred and fifty vibration records were measured for each of the three bearing conditions, i.e., 3750 vibration records in all.

## 5. Experimental Results and Discussion

One of the main goals of this experiment was to detect the weak periodic impulses generated by small defects of the test bearings and then identify the bearing characteristic frequency components using the proposed method. Moreover, the bearing conditions under the motor operating at 100% full-load were presented because the majority of motors spend a higher percentage of time under high-load conditions in many real-life applications.

### 5.1. Bearing Outer-Race Defect Detection

When the motor was operated full-load at a speed of 1741 rpm ($f_{r}$ = 29 Hz), the characteristic frequency of outer-race defect $f_{o}$ is approximately 104 Hz. The time-domain waveform from the acceleration vibration signal generated by a test bearing with an outer-race fault and its corresponding amplitude spectrum are shown in Figure 6. The impulsive phenomena are not obvious in the time domain. From the spectral distributions in Figure 6b, it is apparent that significant energy in frequency band lies between 700 to 800 Hz and the largest spectral amplitude appears at about 780 Hz; however, the magnitude of the expected outer-race characteristic frequency is too small to be identified, as marked $f_{o}$ in Figure 6b, because the low harmonics of the bearing characteristic frequencies are strongly masked by other vibration components and easily buried in background noise.

To verify the effectiveness of the proposed method in bearing fault detection, the FK and the Autogram methods were used to analyze the same experimental bearing vibration signal for comparisons. The plot of the FK for the faulty outer-race bearing signal is shown in Figure 7a, giving optimal demodulation band parameters of the center frequency of 937 Hz and the bandwidth of 375 Hz. As shown in Figure 7b, the corresponding SES based on the selected band is totally dominated by the harmonics of the input shaft speed, about 29 Hz, throughout the spectrum and fails to diagnose the outer-race bearing fault.

Figure 7c shows the Autogram for the faulty outer-race bearing signal, which gives the center frequency of 2250 Hz and the bandwidth of 1500 Hz. The outer-race fault information can be extracted from the Autogram-based combined SES in Figure 7d, but the shaft rotational frequency component about 29 Hz is the largest, and the diagnosis is not particularly clear.

The proposed method was next applied to the same faulty outer-race bearing signal. The TEASK plot is shown in Figure 7e and the maximum value is found with the center frequency of 4500 Hz and the bandwidth of 3000 Hz, at node (1,2). The TEPS of the signal extracted from the node (1,2) is presented in Figure 7f. As shown in the spectrum, the level of noise in the spectrum is drastically reduced and the harmonics of $f_{o}$ are significantly enhanced, which can yield better visual inspection abilities. Moreover, compared with Figure 7b, the TEASK can avoid the interference from the input shaft speed to provide more sensitive information for assessing bearing failure symptoms.

For the faulty outer-race bearing signal under full-load, the proposed method is superior to the other two methods in interference suppression, the SNR enhancement and characteristic frequency components identification. So the proposed method performs better than the FK and the Autogram methods in bearing outer-race defect detection.

### 5.2. Bearing Inner-Race Defect Detection

For this test, the shaft rotational frequency $f_{r}$ is 28.86 Hz when the motor run at 1733 rpm under full-load operation and the characteristic frequency of inner-race defect $f_{i}$ is about 156 Hz.

The time-domain waveform of the acceleration vibration signal form a test bearing with an inner-race fault under full-load condition is shown in Figure 8a. From Figure 8b, the amplitude spectrum presents the largest energy components and does not reveal any inner-race fault characteristic components.

For comparison, both the FK and the Autogram methods were applied to the same bearing faulty inner-race signal. In Figure 9a, the optimal filter indicated by the FK has the center frequency of 5000 Hz and the bandwidth of 2000 Hz according to the highest kurtosis value. The SES of the signal filtered by the optimal filter is depicted in Figure 9b, which does not contain useful diagnostic information. Fault related fault frequencies are nearly overwhelmed by strong noise.

The Autogram was then used to analyse the same inner-race fault signal. The paving of the Autogram is displayed in Figure 9c, where it is indicated that the optimal filter has a center frequency of 3000 Hz and a bandwidth of 6000 Hz at node (0,0). The combined SES of the signal for node (0,0), which is the original inner-race fault signal, is given in Figure 9d. It can be seen in Figure 9d that the Autogram is unable to detect the bearing inner-race fault as the $f_{i}$ frequency component cannot be found in the spectrum.

Finally, the proposed method was applied to the same inner-race fault signal. The plot of the TEASK is shown in Figure 9e and the maximum value of kurtosis is found with the center frequency of 375 Hz and the bandwidth 750 Hz, at node (3,1). The TEPS of the signal extracted from the node (3,1) is presented in Figure 9f, where the inner-race fault characteristic frequency $f_{i}$ is clearly identified.

It is well known that bearing signals are usually modulated by other sources such as shaft frequency. This phenomenon can be observed in Figure 9f that the second largest peak appearing at around 58 Hz (2$f_{r}$) is the second harmonic of the rotational shaft frequency and the third largest peak is the third-order upper-sideband $f_{i}+2f_{r}$. The third-order lower-sideband $f_{i}-2f_{r}$ is suppressed because the relative magnitudes of the sidebands depend on the load distribution between the rolling elements when they strike the inner-race defect. According to [41], the vibrations produced by axial imbalance can be characterized primarily by the second harmonics of the shaft running speed in the spectrum. The peak appearing at 2$f_{r}$ may be caused by the misaligned shaft. Moreover, when a small inner-race defect enters and leaves the load zone once in the revolution, it generates the amplitude modulation [42]. The amplitude modulation generated by multiplying the $f_{i}$ and $f_{r}$ can be represented as
(21)$$\begin{array}{cc} y(t)= & Acos\left(2\pi pf_{i}t\right)\left[1+cos\left(2\pi qf_{r}t\right)\right]\\ = & Acos\left(2\pi pf_{i}t\right)+\frac{A}{2}\left[cos2\pi \left(pf_{i}+qf_{r}\right)t+cos2\pi \left(pf_{i}-qf_{r}\right)t\right] \end{array}$$
where *p* = 1,2,3,⋯, *q* = 1,2,3,⋯ and *A* is the amplitude.

The occurrence of sidebands is evident for the strong non-linear relationship between the shaft rotational frequency and the inner-race fault characteristic frequency. This confirms that the proposed method can successfully detect a small detect in the bearing’s inner-race.

Compared with the FK-based and Autogram-based methods to demodulate the filtered signals using Hilbert square envelope demodulation technique (see Figure 9b,d), the TEPS provides better envelope demodulation ability for the subsequent envelope spectrum analysis to detect and separate different sideband families as shown in Figure 9f.

In contrast to the diagnosis of bearing inner-race fault, it can be clearly seen from Figure 7f that the amplitude modulated side-bands are suppressed around the outer-race frequency when a small defect appears on the outer-race of the bearing. The reason is that the outer-race is restrained from axial and radial movement by a housing that is inherent to the casing and tries to keep in the load zone of the bearing. It should be noted here that the component 2$f_{r}$ does not appear in all the spectra of the outer-race fault by carefully examining all the measured outer-race vibration signals in this experiment. Hence, it is concluded that the non-linearity-related misalignment of the rotating shaft only occurred in the inner-race fault in this study.

From the comparison of experimental results, the proposed method has better demodulation performance and faults extraction ability than the FK-SES and the Autogram-SES in the presence of other vibration interferences.

## 6. Conclusions

This paper proposes a new bearing diagnostic method to determine the fault sensitive frequency bands, where the weak periodic impulses can be accurately revealed and identified in the presence of strong noise. At first, the MODWPT was used as a filter to decompose a signal into different frequency bands and central frequencies. Second, the TEPS of the signals filtered by MODWPT at each node was calculated to enhance the SNR of the spectrum and sharpen the spectral peaks to extract the weak impulses generated by a defective bearing. Third, an adaptive spectral kurtosis was adopted and then computed by combining the overlapped and shifted TEPS segments at each node to construct a two-dimensional color map. From the paving of the TEASK, the sub-band signal with the highest kurtosis is selected as the optimal frequency band for demodulation. In analyzing the simulated signal corrupted by heavy noise, the proposed method has a better ability to detect fault frequencies even though the SNR is very low. Moreover, the proposed method is able to find out the optimal resonant frequency band to locate the center frequency and the bandwidth at different nodes; however, the FK and the Autogram methods used in this study fail to diagnose the same simulated signals under heavy noise. The proposed method has been further tested on real bearing fault vibration signals obtained in the laboratory and the bearing faults are outer-race and inner-race. In terms of demodulation performance, the proposed method has been compared with the FK-based and the Autogram-based demodulation methods. The experimental results demonstrate that the proposed method achieves higher performance and is effective to extract and separate weak fault-related signatures from signals with heavy interferences. The main limitations identified during the presented research are the selection of the mother wavelet function for identifying the given types of faults and the number segments for the computation of the TEASK to determine the optimal frequency band. Although the proposed MODWPT-TEASK approach has been applied in this paper to the diagnosis of the single fault in ball bearing operating in steady state, it can be extended to diagnose compound faults of bearing under non-stationary operating conditions. In addition, the applicability of the TEASK index will be considered for the multi-component amplitude and frequency demodulation.

## Figures and Tables

**Figure 1 sensors-21-06895-f001:**
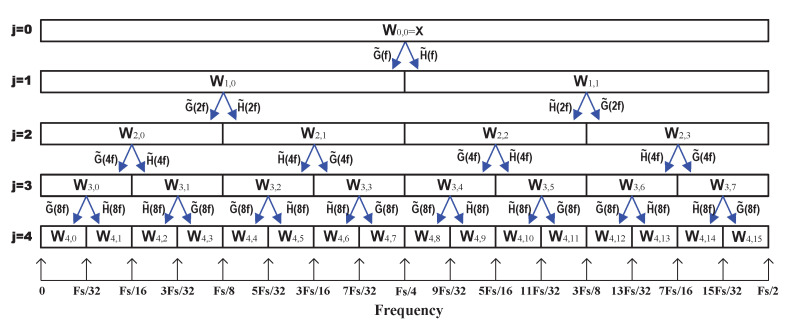
Four levels of the MODWPT decomposition of a time series X.

**Figure 2 sensors-21-06895-f002:**
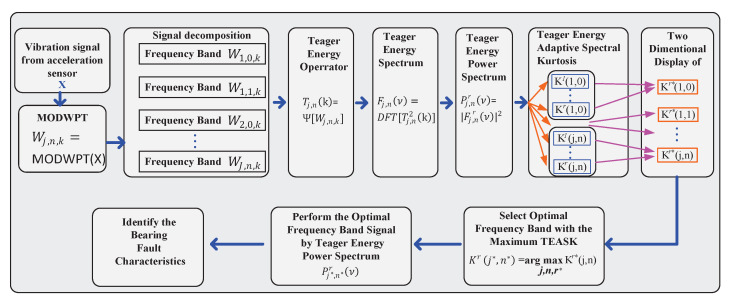
The flowchart of the proposed method for bearing fault detection.

**Figure 3 sensors-21-06895-f003:**
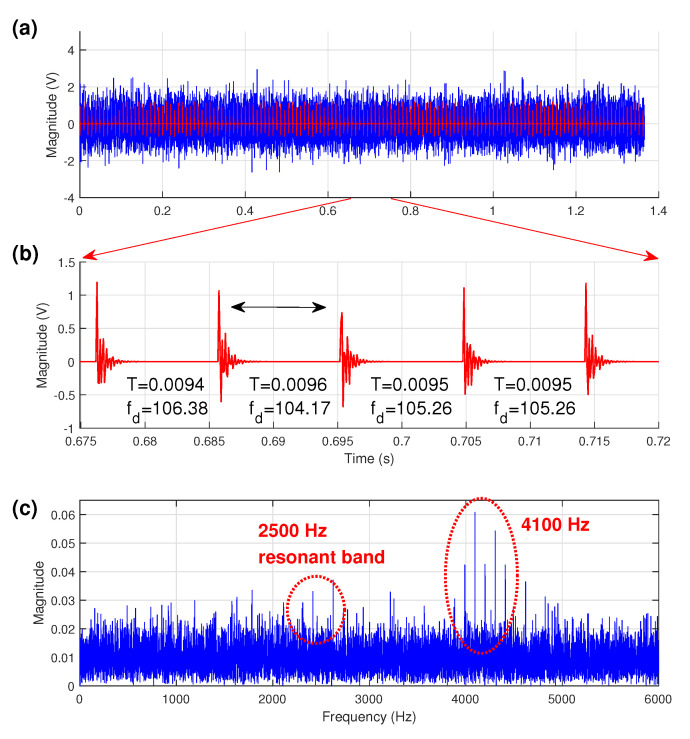
The simulation signal (**a**) time signal with noise (the red waveform represents the uncontaminated signal), (**b**) a part of impulse signal, and its (**c**) amplitude spectrum.

**Figure 4 sensors-21-06895-f004:**
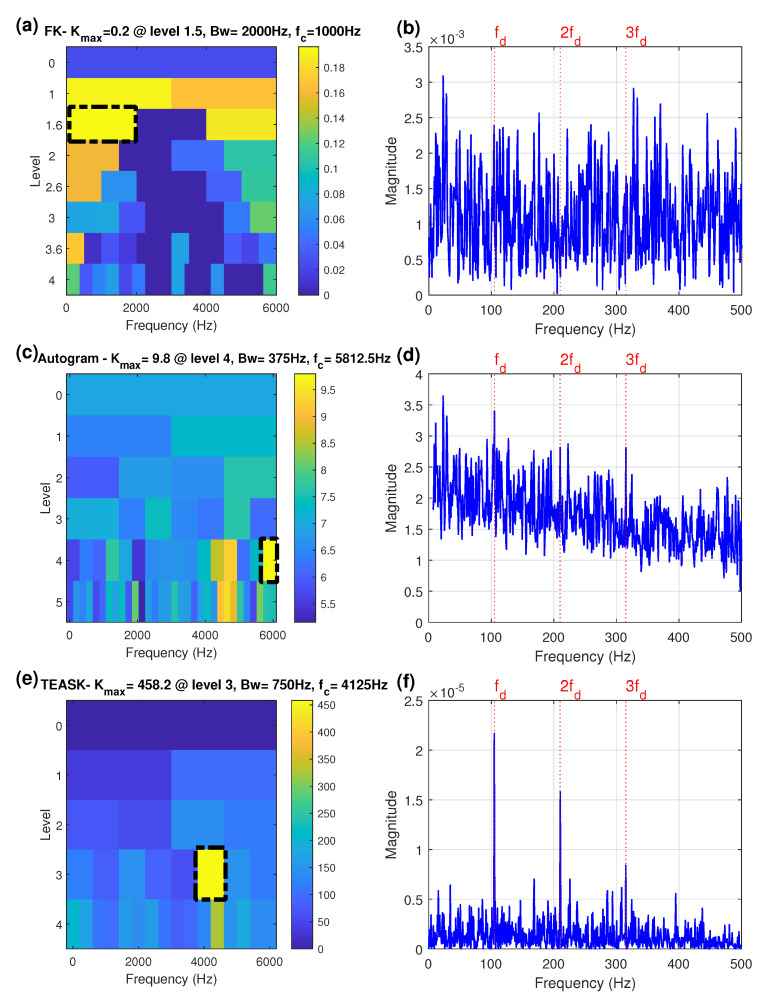
The simulation signal (**a**) the FK, (**b**) the SES obtained by the node with highest kurtosis of the FK, (**c**) the Autogram, (**d**) the combined SES obtained by the node with highest kurtosis of the Autogram, (**e**) the TEASK, (**f**) the TEPS obtained from the node with highest kurtosis of the TEASK. (Red dotted lines: defect frequency component of 105 Hz ($f_{d})$ and its harmonics; black dash-dot rectangle indicates the optimal frequency band).

**Figure 5 sensors-21-06895-f005:**
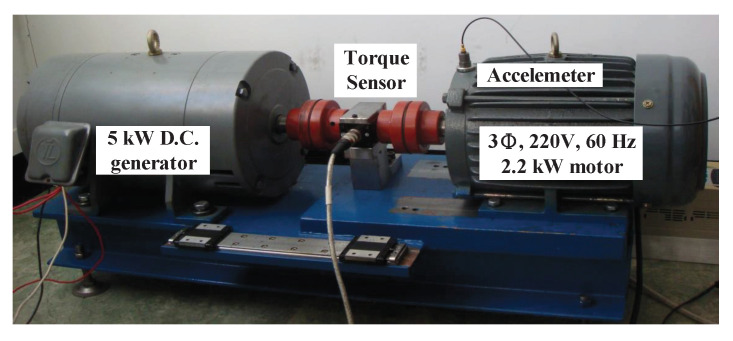
Experimental set-up.

**Figure 6 sensors-21-06895-f006:**
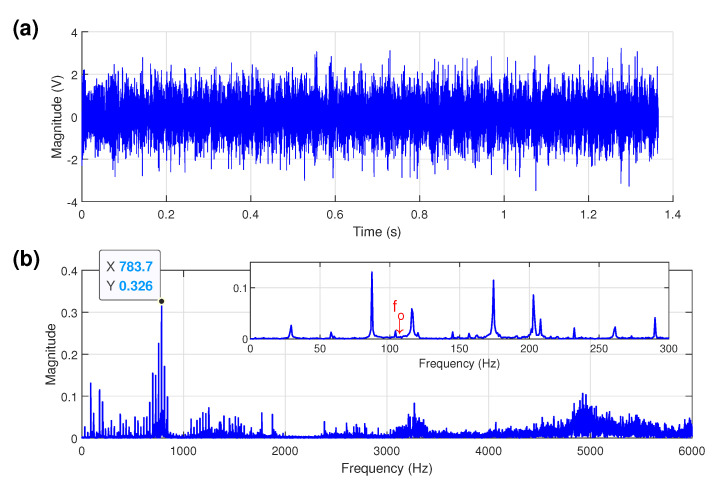
Faulty outer-race bearing: (**a**) time signal; (**b**) amplitude spectrum.

**Figure 7 sensors-21-06895-f007:**
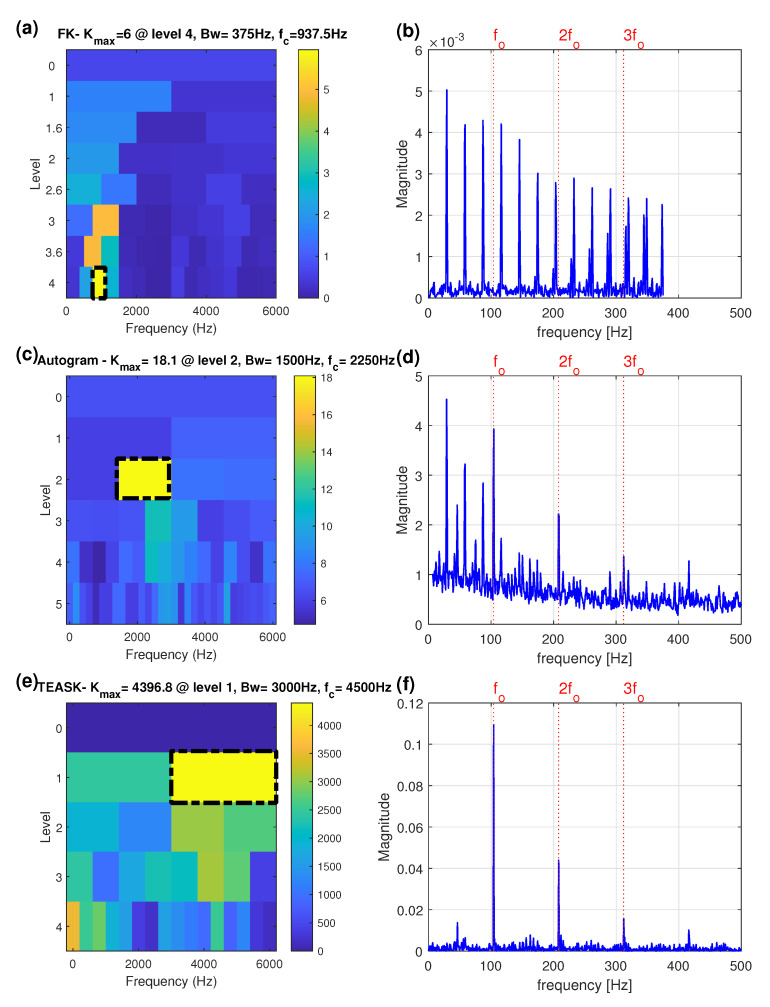
Faulty outer-race bearing under full-load: (**a**) the FK; (**b**) the SES obtained by the node with highest kurtosis of the FK; (**c**) the Autogram; (**d**) the combined SES obtained by the node with highest kurtosis of the Autogram; (**e**) the TEASK; (**f**) the TEPS obtained from the node with highest kurtosis of the TEASK. (Red dotted lines: outer-race frequency component of 104 Hz ($f_{o})$ and its harmonics; black dash-dot rectangle indicates the optimal frequency band).

**Figure 8 sensors-21-06895-f008:**
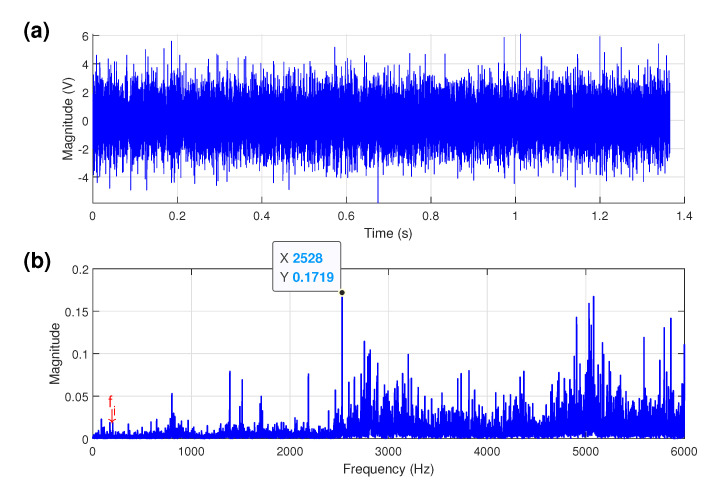
Faulty inner-race bearing: (**a**) time signal; (**b**) amplitude spectrum.

**Figure 9 sensors-21-06895-f009:**
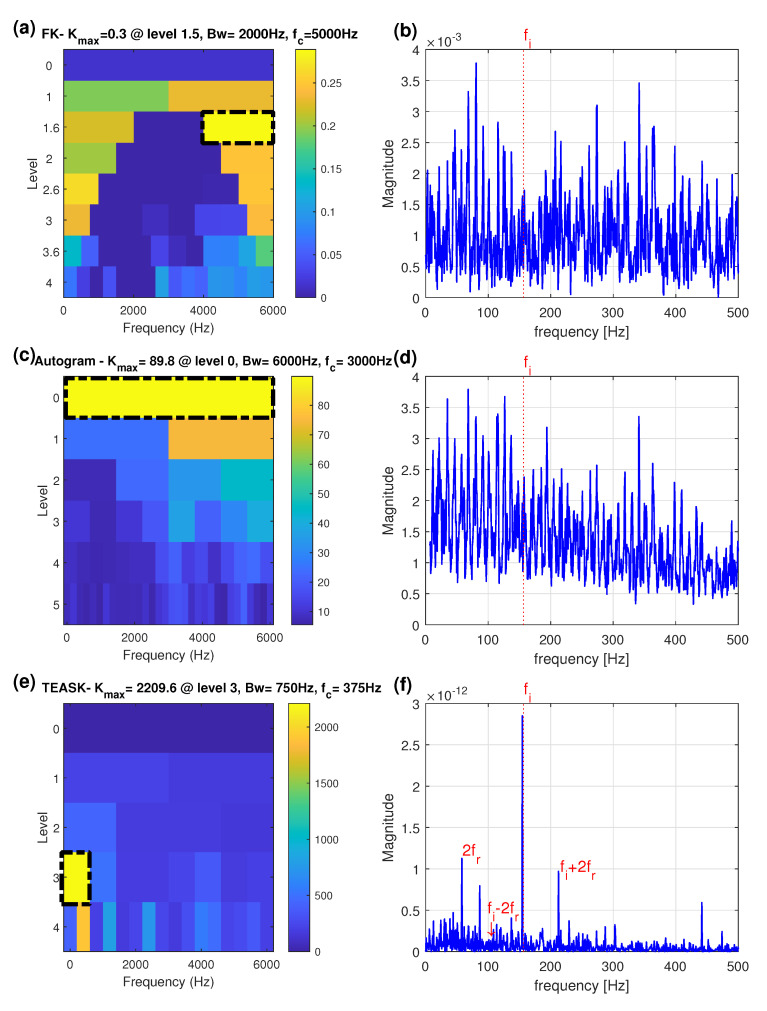
Faulty inner-race bearing: (**a**) the FK; (**b**) the SES obtained by the node with highest kurtosis of the FK; (**c**) the Autogram; (**d**) the combined SES obtained by the node with highest kurtosis of the Autogram; (**e**) the TEASK; (**f**) the TEPS obtained from the node with highest kurtosis of the TEASK. (The black dash-dot rectangle indicates the optimal frequency band).

**Table 1 sensors-21-06895-t001:** Simulated bearing fault parameters.

Parameters	β	$f_{d}$ (Hz)	$A_{1}$	$f_{1}$ (Hz)	$A_{2}$	$f_{2}$ (Hz)	$\tau_{r}$
Values	1000	105	0.4	2500	1	4100	1.5%

**Table 2 sensors-21-06895-t002:** Parameters of bearings.

Type	Bore	Outer	Width	Number	Ball	Pitch	$f_{o}$	$f_{i}$
	Diameter	Diameter		of Balls	Diameter	Diameter		
	(mm)	(mm)	(mm)		(mm)	(mm)	(Hz)	(Hz)
6206	30	62	16	9	9.5	46.46	3.58$f_{r}$	5.42$f_{r}$

Where $f_{o}$ is the outer-race frequency, $f_{i}$ is the inner-race frequency, and $f_{r}$ is the shaft rotational frequency.

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
