# Peer review of "The Detection of Motor Bearing Fault with Maximal Overlap Discrete Wavelet Packet Transform and Teager Energy Adaptive Spectral Kurtosis"

_sensors, 2021, doi:10.3390/s21206895_

Round 1

Reviewer 1 Report

This paper proposed an effective bearing diagnostic method that integrate the merits of MODWPT, adaptive SK and the TEPS to reveal the weak periodic impulses in the presence of strong noise. Comments about the paper are as follow:  

  • The abstract section is too long and needs to be simplified.
  • Modify Figure 2 to make it look better and check the case problem about the (j∗, n∗). What’s the meaning about the “Alter the operator”?
  • In Figure 7(f), the proposed method clearly identified the outer-race fault characteristic frequency 105 Hz and its harmonics without any other interference; but in Figure 9(f), other frequency information such as the second harmonic of shaft frequency can be observed, how to explain this phenomenon in the paper?
  • Outer race fault and inner race fault were verified in the experiment part, but how about the ball fault? What if the outer race fault and inner race fault were introduced simultaneously? What if there are other working conditions? The experimental content should be extended to make the proposed method more convincing.
  • In Line 328 to 334, the choices of hyper-parameters such as “db12” wavelet and window size are based on which criterions? They should be mentioned so that readers can know how to transfer the proposed method to other diagnostic problems.

Author Response

Manuscript ID: sensors-1355054

Title: The detection of motor bearing fault with maximal overlap discrete wavelet packet transform and Teager energy adaptive spectral kurtosis

--------------------------------------------------------------------------

Responses to Reviewer #1 (Reviewer’s comments are shown in Italic)

I express my sincere appreciation to the reviewer for providing the valuable comments and suggestions. These comments are very constructive, and will help us to improve the quality of the manuscript, specifically in terms of clarifying our methodology. I address the reviewer’s concerns in this letter, and corresponding changes have made to improve the original submission.

Comment 1: The abstract section is too long and needs to be simplified.

Response: I thank the reviewer for this excellent suggestion. The abstract has been shortened. I have reduced the length of the abstract from 276 to 166 in the revised manuscript. These sentences in the Abstract of the revised manuscript read as:

Motor bearings are one of the most critical components in rotating machinery. Envelope demodulation analysis has been widely used to demodulate bearing vibration signals to extract bearing defect frequency components but one of the main challenges is to accurately locate the major fault-induced frequency band with a high signal-to-noise ratio (SNR) for demodulation. Hence, an enhanced fault detection method combining the maximal overlap discrete wavelet packet transform (MODWPT) and the Teager energy adaptive spectral kurtosis (TEASK) denoising algorithms is proposed for identifying the weak periodic impulses. The Teager energy power spectrum (TEPS) defines the sparse representation of the filtered signals of the MODWPT in the frequency domain via the Teager energy operator (TEO). And the TEASK helps determine the most informative frequency band for demodulation. The methodology is compared in terms of performance with the fast Kurtogram and the Autogram methods. The simulation and practical application examples have shown that the proposed MODWPT-TEASK method outperforms the above two methods in diagnosing defects of motor bearings.

Comment 2: Modify Figure 2 to make it look better and check the case problem about the (j, n). What’s the meaning about the “Alter the operator”?

Response: I thank the reviewer for the kind comment and careful check of our Figures. I have modified the Figure 2 and the (j*,n*) has been revised accordingly.

In the original submission, “Alter the operator” in Figure 2 was to inform the maintenance personnel in making maintenance decisions. Now I think “Alter the operator” is superfluous and should be deleted to make the flowchart more legible in the revised manuscript.

Figure 2 (the original submission)

Please see the attached file  called author-coverletter-14037698.v1.pdf.

Figure 2 (the revised manuscript)

Please see the attached file  called author-coverletter-14037698.v1.pdf.

Comment 3: In Figure 7(f), the proposed method clearly identified the outer-race fault characteristic frequency 105 Hz and its harmonics without any other interference; but in Figure 9(f), other frequency information such as the second harmonic of shaft frequency can be observed, how to explain this phenomenon in the paper?.

Response: I thank the reviewer for pointing this important issue out. In the revised manuscript, this phenomenon have been explained (page 15, lines 452-454) with two sentences and one new reference ([41]). These sentences read as: “According to [41], the vibrations produced by axial imbalance can be characterized primarily by the second harmonics of the shaft running speed in the spectrum. The peak appearing at 2fr may be caused by the misaligned shaft.

Comment 4: Outer race fault and inner race fault were verified in the experiment part, but how about the ball fault? What if the outer race fault and inner race fault were introduced simultaneously? What if there are other working conditions? The experimental content should be extended to make the proposed method more convincing.

Response: I thank the reviewer for this excellent suggestion. I agree that adding the ball fault and compound fault data would certainly be of interest. Unfortunately, at the time the experiments were conducted, the ball fault and compound fault conditions were not taken into consideration. The problem will be tackled in the future study using one single ball fault (BF) and compound faults (OF -IF, OF-BF, IF-BF, and OF-IF -BF), where OF and IF represent outer race fault and inner race fault, respectively.

In the revised manuscript, I have added four sentences (pages 11-12, lines 368-372 and page 12, line374-376) to extend the experimental content in the Experimental apparatus. These sentences read as: “Each bearing was tested for the motor being unloaded (0%), 25%, 50%, 75% loaded and fully loaded (100%). Data set of the bearing was acquired from the experimental system under three different operating conditions: (1) normal condition, (2) with outer-race defect, and (3) with inner-race defect. Therefore, the vibration signals were acquired under different operating loads and bearing conditions.” And “Two hundred and fifty vibration records were measured for each of the three bearing conditions, i.e. 3750 vibration records in all.

In the revised manuscript, I also have added one sentence (page 12, lines 380-383) in the Experimental results and discussion to explain why the only full-load condition was used. This sentence read as: ” Moreover, the bearing conditions under the motor operating at 100% full-load were presented because the majority of motors spend a higher percentage of time under high-load conditions in many real-life applications.

The ball fault and the outer race fault and inner race fault are explained in the Conclusions with one sentence (page 17, lines 502-505) in the revised manuscript. This sentence read as: ” Although the proposed MODWPT-TEASK approach has been applied in this paper to the diagnosis of the single fault in ball bearing operating in steady state, it can be extended to diagnose compound faults of bearing under non-stationary operating conditions.

Comment 5: In Line 328 to 334, the choices of hyper-parameters such as “db12” wavelet and window size are based on which criterions? They should be mentioned so that readers can know how to transfer the proposed method to other diagnostic problems.

Response: I thank the reviewer for the suggestions. In the revised manuscript, three sentences (page 11, lines 333-337) have been added to explain the use of db12 in the Simulation analysis with one new reference ([38]). This sentence read as: ” Before performing the MODWPT on the signal, the most crucial step is the selection of the mother wavelet function as well as the decomposition level of the signal. Among orthogonal wavelets, Daubechies (DB) wavelets have been widely used because they match the transient components in vibration signals. Daubechies wavelets (db12) used in [1538] were adopted in this study to decompose the signal and four decomposition levels was used in this investigation.

In the revised manuscript, I have added two sentences to explain the use of window size with two new reference ([39]、[40]). This sentence read as: ”The purpose of overlapping segments is to decrease the TEPS estimate variance and then improve the SNR of a spectrum. According to p. 144 of [39] and p. 347 of [40], a 50 % overlap can lead to a very efficient computation with the FFT algorithm and the maximum reduction of the variance obtained by segmentation.

Reviewer 2 Report

Dear Author,

Based on the first-round review of the manuscript entitled The detection of motor bearing fault with maximal overlap discrete wavelet packet transform and Teager energy adaptive spectral kurtosis, the reviewer has the following comments:

  1. please more explain the contribution of the manuscript in the introduction.
  2.  We have various data-driven and model-based approaches for bearing fault diagnosis. What are the advantages of the proposed method compared with the data-driven, model-based, and hybrid approaches?
  3. Regarding the manuscript, the proposed method is useful for noisy signals. So, how you can validate the robustness of the proposed method?
  4.  Is your technique reliable? if yes, how you can validate it?
  5. Is it possible to applied the proposed method for accoustic emmission signals?
  6. What is the limitation of the proposed method? Please explain it in the conclusion. 
  7. Please explain about the future work in conclusion.

Regards,

Author Response

Manuscript ID: sensors-1355054

Title: The detection of motor bearing fault with maximal overlap discrete wavelet packet transform and Teager energy adaptive spectral kurtosis

--------------------------------------------------------------------------

Responses to Reviewer #2 (Reviewer’s comments are shown in Italic)

I express my sincere appreciation to the reviewer for providing the valuable comments and suggestions. These comments are very constructive, and will help us to improve the quality of the manuscript, specifically in terms of clarifying our methodology. I address the reviewer’s concerns in this letter, and corresponding changes have made to improve the original submission.

Comment 1: please more explain the contribution of the manuscript in the introduction.

Response: I thank the reviewer for reminding these important issues. In response to the reviewer's suggestion, one sentence (page 4, lines 148-151) has been added to explain the contribution of this study in the Introduction of the revised manuscript. This sentence read as: ”By integrating the MODWPT with TEPS, a new bearing detection tool, named the MODWPT-TEASK, for the analysis of vibration signals resulting from bearings with localized defects is presented, which can be used for the diagnosis of single, simultaneous and early weak fault.

Comment 2: We have various data-driven and model-based approaches for bearing fault diagnosis. What are the advantages of the proposed method compared with the data-driven, model-based, and hybrid approaches?

Response: I thank the reviewer for raising an interesting point, but our study mainly focused on signal-based fault detection approaches. Compared with model-based approaches, the proposed approach does not need to build an explicit mathematical model to describe the physics of rolling element bearing. Because real-world component-system physics is often too complex and stochastic to build models. Moreover, the proposed approach can extract essential information with less knowledge about input signals. Although hybrid approaches can integrate the advantages of model-based and data-driven approaches, it is still difficult to select, combine, and optimize the parameters of various procedures.

Comment 3: Regarding the manuscript, the proposed method is useful for noisy signals. So, how you can validate the robustness of the proposed method?

Response: I thank the reviewer for the positive feedback. Our explanations are that two stages in development of the proposed approach are used to validate its effectiveness. At the first stage, we verified the performance of the proposed approach with simulated signals under various SNRs to test whether the proposed method is insensitive to the signal uncertainties. The development stage is verified by the experimental data to validate the effectiveness of the proposed method.

Comment 4: Is your technique reliable? if yes, how you can validate it?

Response: I thank the reviewer for raising this important point. Reliability test are concerned with repeatability or consistency. Although the experimental data analyzed by the proposed approach cannot be overall level of reliability, the test-retest reliability for experimental bearing data of the outer-race fault and the inner-race fault under full-load is high. But how to quantify the level of reliability is not within the scope of this study.

Comment 5: Is it possible to applied the proposed method for accoustic emmission signals?

Response: While I agree that the application of acoustic emission technology for fault diagnosis of bearings is an interesting topic, the focus of this study is only based on vibration signals to extract bearing fault features. Since acoustic emission data were not collected initially, we are unable to perform the request analysis. I will seriously considered your suggestion as our future direction.

Comment 6: What is the limitation of the proposed method? Please explain it in the conclusion.

Response: I thank the reviewer for this excellent suggestion. In response to the reviewer's suggestion, one sentence (page 17, lines 498-502) has been added to explain the limitation of the proposed method in the Conclusion of the revised manuscript. This sentence read as: ” The main limitations identified during the presented research are: the method requires the user to select the mother wavelet function which can influence the ability to identify the given types of faults; the choices of the window size and the overlapping percentage for the computation of the TEASK also influence the determination of the optimal frequency band for demodulation.

Comment 7: Please explain about the future work in conclusion.

Response: I thank the reviewer for this excellent suggestion. In response to the reviewer's suggestion, two sentences (page 17, lines 502-505) have been added to explain the future work in the Conclusion of the revised manuscript. This sentence read as: ” Although the proposed MODWPT-TEASK approach has been applied in this paper to the diagnosis of the single fault in ball bearing operating in steady state, it can be extended to diagnose compound faults of bearing under non-stationary operating conditions. In addition, the applicability of the TEASK index will be considered for the multi-component amplitude and frequency demodulation.

Reviewer 3 Report

In the article, the author proposes a novel method of identifying weak periodic pulses, combining the maximal overlap discrete wavelet packet transform (MODWPT) and the Teager energy adaptive spectral kurtosis (TEASK) denoising algorithms. The proposed method has been tested with simulation signals and then with real bearing fault vibration signals obtained in the laboratory. In terms of demodulation performance, the proposed method has been compared with the FK-based and the Autogram-based demodulation methods. 

Author Response

Manuscript ID: sensors-1355054

Title: The detection of motor bearing fault with maximal overlap discrete wavelet packet transform and Teager energy adaptive spectral kurtosis

-------------------------------------------------------------------------------------------------------

Responses to Reviewer #3

In the article, the author proposes a novel method of identifying weak periodic pulses, combining the maximal overlap discrete wavelet packet transform (MODWPT) and the Teager energy adaptive spectral kurtosis (TEASK) denoising algorithms. The proposed method has been tested with simulation signals and then with real bearing fault vibration signals obtained in the laboratory. In terms of demodulation performance, the proposed method has been compared with the FK-based and the Autogram-based demodulation methods.

Response: I would like to thank the reviewer for your time and effort in reviewing this paper.

Round 2

Reviewer 1 Report

The authors have carefully responded the comments, but there are still some  deficiencies to be fixed.

In Figure 2, the fault diagnosis flowchart should not include the "End" step, ellipsis should not be drawn in term of dotted line.

In line 452-454, I had noticed that the author have explained the harmonic phenomenon for Figure 9, but I wonder why Figure 7 didn't have this phenomenon, the author may briefly mention whether this only occurs in inner-race fault or not. 

 In line 143, the paper didn't discuss the compound fault issue, so " which can be used for the diagnosis of single, simultaneous and early weak fault." is not reliable.

In line 328 to 334, the author have mentioned the details for the hyper-parameter setups, but there is no need to excessively emphasize some common knowledges, line 333-337 and 490-494 may be modified or simplified for making better writing.

Author Response

Comment 1: In Figure 2, the fault diagnosis flowchart should not include the "End" step, ellipsis should not be drawn in term of dotted line.

Response: I thank the reviewer again for the kind comment and careful check of our Figures. In response to the reviewer’s suggestion, I have modified the Figure 2.

Figure 2, please see the attached file

Comment 2: In line 452-454, I had noticed that the author have explained the harmonic phenomenon for Figure 9, but I wonder why Figure 7 didn't have this phenomenon, the author may briefly mention whether this only occurs in inner-race fault or not.

Response: I thank the reviewer for allowing me to explain more regarding this issue. In the 2nd revised revision, I have added two sentences (page 15, lines 465-469) in the Experimental results and discussion to explain this phenomenon. These sentences read as: “It should be noted here that the component 2fr does not appear in all the spectra of the outer-race fault by carefully examining all the measured outer-race vibration signals in this experiment. Hence, it is concluded that the non-linearity-related misalignment of the rotating shaft only occurred in the inner-race fault in this study.

Comment 3: In line 143, the paper didn't discuss the compound fault issue, so " which can be used for the diagnosis of single, simultaneous and early weak fault." is not reliable.

Response: I thank the reviewer for pointing the careful usage of word out. In the 2nd revised revision, I have deleted the word of “simultaneous”. The sentence (page 3, lines 140-143) read as: “By integrating the MODWPT with TEPS, a new bearing detection tool, named the MODWPT-TEASK, for the analysis of vibration signals resulting from bearings with localized defects is presented, which can be used for the diagnosis of single and early weak fault.

Comment 4: In line 328 to 334, the author have mentioned the details for the hyper-parameter setups, but there is no need to excessively emphasize some common knowledges, line 333-337 and 490-494 may be modified or simplified for making better writing.

Response: I thank the reviewer for this excellent suggestion. In the 2nd revised revision, I have modified the sentence (page 11, lines 332-335) and read as: “At the same time, the window size used in the TEPS was set to half of the original signal with an overlapping of 50% (see p. 144 of [39] and p. 347 of [40]) which is half the window length for decreasing the TEPS estimate variance and then improving the SNR of the TEPS.

Lines 332-337 in the 1st revised revision,

At the same time, the window size used in the TEPS was set to half of the original signal with an overlapping of 50% which is half the window length. The purpose of overlapping segments is to decrease the TEPS estimate variance and then improve the SNR of a spectrum. According to p.144 of [39] and p.347 of [40], a 50 % overlap can lead to a very efficient computation with the FFT algorithm and the maximum reduction of the variance obtained by segmentation.

Moreover, I also have modified the sentence (page 17, lines 499-502) in the 2nd revised revision and read as: “The main limitations identified during the presented research are the selection of the mother wavelet function for identifying the given types of faults and the number segments for the computation of the TEASK to determine the optimal frequency band.

Lines 490-494 in the 1st revised revision,

The main limitations identified during the presented research are: the method requires the user to select the mother wavelet function which can influence the ability to identify the given types of faults; the choices of the window size and the overlapping percentage for the computation of the TEASK also influence the determination of the optimal frequency band for demodulation.

I thank the reviewer for excellent suggestions. I have attached the 2nd revised revision (see pages 4-22/40 of the attached pdf), where the revised portions are indicated by blue fond and the deleted portions are indicated by red strike-through font. At the same time, a clear version is also provided (see pages 23-40/40 of the attached pdf). Finally, I express my sincere appreciation to the reviewer for your extremely careful reading and checking our manuscript. Your comments are indispensable to improve the quality of the manuscript. The role you play is like my virtual supervisor! I really appreciate it and learn a lot from your comments you have given. Once again, thank you so much for your time and effort in reviewing this manuscript.

Reviewer 2 Report

Dear Authors,

Based on the 2nd round review of the manuscript entitled The detection of motor bearing fault with maximal overlap discrete wavelet packet transform and Teager energy adaptive spectral kurtosis, it can be accepted for further processing.

Regards,

Author Response

Manuscript ID: sensors-1355054

Title: The detection of motor bearing fault with maximal overlap discrete wavelet packet transform and Teager energy adaptive spectral kurtosis

-------------------------------------------------------------------------------------------------------

Based on the 2nd round review of the manuscript entitled The detection of motor bearing fault with maximal overlap discrete wavelet packet transform and Teager energy adaptive spectral kurtosis, it can be accepted for further processing.

Response: Thank you very much for your careful reading of our revised manuscript and I am delighted that you recommend for acceptance of this manuscript.
